# Transformer Instability in Long Sequence Training: The Underestimated Role of Short-Range Dependencies

## Abstract

Transformer language models have driven remarkable progress across diverse fields, including natural language processing, speech processing, and computer vision. However, despite extensive research, transformers remain prone to training instability on long sequences, often manifesting as sudden spikes or divergence in the training loss during a run.

In this work, we identify a key source of this instability: self-attention's limited capacity to capture short-range dependencies – particularly in tasks such as language modeling, where most tokens depend heavily on their immediate neighbors. This limitation leads to rapid growth of the self-attention's logits during long-sequence training, ultimately destabilizing optimization.

To address this, we propose augmenting the standard architecture with several local (short-range) attention heads alongside the full (long-range) attention heads. The local heads explicitly capture short-range dependencies, while the full heads preserve long-range context. This composed self-attention – termed Long Short-attention (LS-attention) – stabilizes training by mitigating logit explosion. Across a wide range of experiments, we demonstrate that long-sequence training triggers logit explosion for multi-head self-attention (MHSA), whereas LS-attention effectively prevents it. Additionally, LS-attention makes transformer models more efficient, reducing inference latency by up to $44\%$ compared to equivalent state-of-the-art MHSA implementations.

## 1 Introduction

Transformer language models have become the backbone of modern machine learning systems, achieving remarkable success across diverse domains such as natural language processing (Vaswani et al. (2017); Devlin et al. (2019); Radford et al. (2018; 2019)), computer vision (Chen et al. (2020); Yu et al. (2022); Pippi et al. (2025); Chang et al. (2022)), and speech (Baevski et al. (2020); Hsu et al. (2021); Ao et al. (2022); Gulati et al. (2020)). These models have enabled state-of-the-art results in applications like machine translation, document summarization, code generation, image captioning, and multimodal reasoning.

Despite their immense success, transformer language models often exhibit training instability, particularly during large-scale pretraining or when processing long sequences (Molybog et al. (2023); Chowdhery et al. (2023); Li et al. (2022); Wortsman et al. (2024); Zhai et al. (2023); Dehghani et al. (2023); Nishida et al. (2024); Wang et al. (2025); Kedia et al. (2024)). This instability typically manifests as spikes or divergence in the training loss. Several explanations and solutions for this training instability have been proposed in the literature. For instance, Liu et al. (2020) attribute instability to the amplification of small parameter perturbations due to reliance on the residual branch. Others, such as Molybog et al. (2023), implicate the Adam optimizer (Kingma & Ba (2015)) as a contributing factor. The use of long sequences during training has also been linked to instability, prompting strategies like progressive sequence length increase during training (Li et al. (2022; 2021)). Several studies, such as Wortsman et al. (2024); Zhai et al. (2023); Dehghani et al. (2023); Kedia et al. (2024), associate the issue with logit explosion and propose normalization techniques (e.g., QK-norm Henry et al. (2020)) to stabilize training, though the root cause of the explosion remains

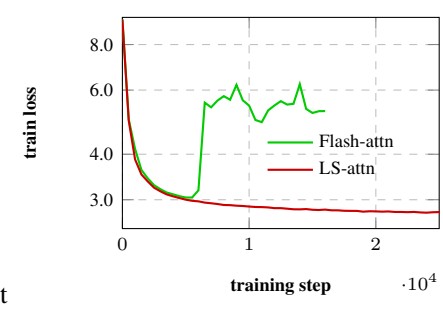
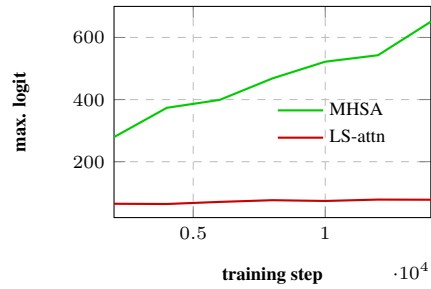

(a) Plot of training loss over training steps.

(b) Plot of maximum absolute logit from attention operations over training steps.

Figure 1: Illustration of training instability and logit explosion in Flash-attention, a state-of-the-art implementation of MHSA. The left plots show that the training loss of an autoregressive transformer with Flash-attention begins to diverge after a certain number of steps for longer sequences ($n = 2K$), whereas the same model with LS-attention (incorporating both local and full attention heads) remains stable. The right plots compare the maximum absolute logits of MHSA and LS-attention during training. LS-attention mitigates logit explosion by reducing the maximum logit magnitude to less than one-eighth of that in MHSA.

unclear. Nishida et al. (2024) identify norm imbalance among parameters as a source of instability and introduce reparameterization methods to address it. Additional techniques such as learning rate warm-up, weight decay, and $\mu$Param (Yang et al. (2022)) have also been explored. However, a clear understanding of the underlying causes – particularly those stemming from the behavior of the self-attention mechanism – and their effective mitigation remains an active area of research.

**Cause of Instability:**  Although several studies (e.g., Wortsman et al. (2024); Zhai et al. (2023); Dehghani et al. (2023); Kedia et al. (2024)) have identified the explosion of logits in self-attention as a key contributor to training instability, the underlying cause of this phenomenon remains largely unexplained. In this work, we attribute the logit explosion to self-attention's limited capacity to model local or short-range dependencies – especially in tasks such as natural language processing, where almost every token typically relies heavily on its neighboring tokens. To elaborate, let $X = [\mathbf{x}_0, \ldots, \mathbf{x}_{n-1}]^T \in \mathbb{R}^{n \times d}$ represents a sequence of $n$ input tokens. The self-attention mechanism transforms $X$ into new representations $Y = [\mathbf{y}_0, \ldots, \mathbf{y}_{n-1}]^T \in \mathbb{R}^{n \times d}$, computed as:

$$Y = PXW_v,$$

where $W_v \in \mathbb{R}^{d \times d}$ is a trainable weight matrix, and $P \in \mathbb{R}^{n \times n}$ is the attention matrix encoding the token dependencies. Each row of $P$ is a probability distribution, where a high $P[i,j]$ implies that the representation $\mathbf{y}_i$ strongly incorporates information from $\mathbf{x}_j$. The attention matrix is computed via: $P = \text{softmax}(S) = \text{softmax}(QK^T) = \text{softmax}(XW_Q W_K^T X^T)$[1], where $Q, K \in \mathbb{R}^{n \times d}$ are the query and key matrices, respectively, and $S \in \mathbb{R}^{n \times n}$ contains the pre-softmax logits. To model arbitrary dependencies between $n$ tokens, the attention matrix $P$ ideally requires $O(n^2)$ parameters. However, because $P$ is derived from the product of two $n \times d$ matrices, its number of parameters remains $O(nd)$. When $n \gg d$, this becomes a low-rank bottleneck (Bhojanapalli et al. (2020)). In tasks where all tokens depends on a small set of "keyword" tokens, the attention matrix becomes low-rank, which is well repressented by $O(nd)$ parameters. However, in tasks requiring dense local dependencies – where nearly every token depends on its immediate neighbors – the attention matrix must be effectively high-rank (as shown in Figure 2a). The difficulty of high rank attention matrix using only $O(nd)$ parameters forces the model to compensate by inflating the logits $S$ when $d \ll n$, leading to training instability in these scenarios.

**The Solution:**  The key idea behind our approach to mitigating logit explosion stems from the observation that local dependencies typically span only a small window around each token. As a result, they can be effectively captured using $O(nl)$ parameters, where $l \ll n$ denotes the local

---

[1]Without loss of generality, we ingore the logit scaling factor for simplicity.

window size. In contrast, full attention attempts to model interactions between all pairs of $n$ tokens in the input sequence, requiring the representation of $O(n^2)$ attention weights. This demand often exceeds the expressive capacity of the attention mechanism, since its parameterization is limited to $O(nd)$. A sliding-window local attention mechanism, which restricts each query token's attention span to a small neighborhood of $l'$ tokens ($l' \ll n$), reduces the number of attention scores to be represented to $O(nl')$, making it more compatible with the available parameterization. Local attention is therefore more effective than full attention for capturing dense short-range dependencies. However, local attention alone is insufficient for modeling long-range dependencies, which remain essential for strong performance for many tasks. To meet both needs, we propose using both local (short-range) and full (long-range) attention heads. This composed attention, referred to as LS-attention, enables transformer models to effectively capture both short- and long-range dependencies while reducing the risk of logit explosion during training (as illustrated in Figure 1).

**Efficiency of The Solution:** In addition to improving training stability, LS-attention offers computational efficiency during both training and inference. For longer sequences, the computational overhead of a transformer model is dominated by the MHSA module, which uses full attention heads with quadratic computational complexity in the sequence length $n$. In contrast, a local attention head with attention span $l \ll n$ exhibits nearly linear complexity with respect to $n$. In practice, we find that LS-attention, with only a few full attention heads and the remaining heads as local attention, performs very well, which reduces both training and inference time significantly. In our experiments, we found that a model using LS-attention was up to $44\%$ more efficient during inference compared to one using Flash-attention (Dao et al. (2022); Dao (2024)), the state-of-the-art efficient implementation of MHSA, on longer sequences.

**Summary of Contributions:** The contributions of this work are summarized as follows:

- We identify a key limitation of self-attention: its inability to effectively model dense local dependencies in long sequences. This limitation can lead to logit explosion during long-sequence training, contributing to training instability in transformer models, particularly for tasks such as language modeling.

- To address the above limitation, we propose to compose local (short-range) and full (long-range) attention heads. The composed self-attention, referred to as Long Short-attention (LS-attention), mitigates the logit explosion and stabilizes the training. Through extensive experimentation, we validate that long sequence training leads to logit explosion in MHSA while LS-attention mitigates it.

- We have also investigated the applicability of other structured self-attention mechanisms and training stabilization methods for stabilizing long-sequence training.

## 2 RELATED WORKS

**Low-Rank Bottleneck of Self-Attention** Earler, Bhojanapalli et al. (2020) identified a low-rank bottleneck in the self-attention layer, showing that it may not represent all possible attention matrices P when the embedding dimension $d < n$. To address this, they proposed setting the head dimension to $n$. However, this strategy becomes impractical for large $n$, as it increases the computational complexity of the MHSA operation to $O(Hn^3)$, where $H$ is the number of heads. In contrast, our work identifies a specific scenario – the presence of dense local dependencies – where this low-rank bottleneck leads to critical training instability in transformer networks. Based on this observation, we propose an efficient solution that not only resolves the instability but also improves computational efficiency on long sequences.

**Structured Self-Attention** Various structured self-attention mechanisms have been extensively explored to mitigate the quadratic computational complexity of vanilla self-attention. For example, prior works, such as Child et al. (2019); Beltagy et al. (2020); Zaheer et al. (2020); Jiang et al. (2024); Guo et al. (2019), have proposed replacing full self-attention with combinations of sparse self-attentions – such as local, global, and dilated attention – to overcome the quadratic complexity barrier and improve model efficiency on long sequences. In contrast, our work identifies a key limitation of full attention in capturing dense local dependencies and demonstrates that structured

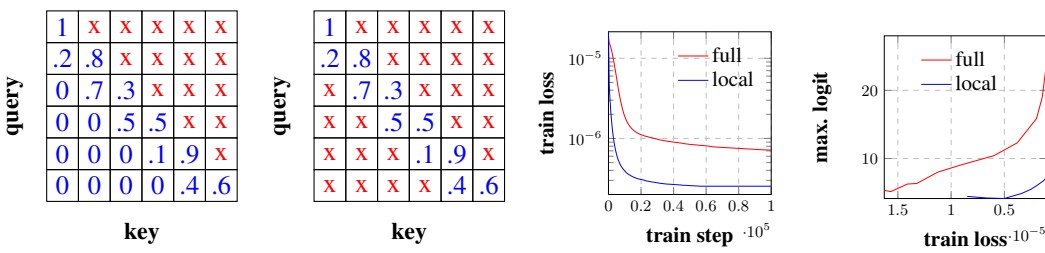

(a) Attention pattern for (causally) full attention.

(b) Attention pattern for local attention.

(c) Training curve for syn. experiment.

(d) Maximum logit vs. train loss on syn. experiment.

Figure 2: Comparison of representing dense local dependencies by local and full attention. (a) Full attention attempts to represent $O(n^2)$ attention scores (shown in blue) using only $O(nd)$ degrees of freedom. (b) Local attention focuses on $O(nl')$ attention scores, where $l' \ll n$, making it a better fit for the available $O(nd)$ capacity. (c) In a synthetic dense local dependency learning task, local attention achieves lower training loss. (d) Local attention requires lower logit values than full attention to achieve the same training loss.

self-attention can be used to address this limitation. Thus, our contribution is to develop efficient self-attention that enhances – rather than compromises – the representational power of full attention.

## 3 IMPACT OF DENSE LOCAL DEPENDENCIES ON LOGIT EXPLOSION IN LONG-SEQUENCE TRAINING

In this section, we analyze the ability of (full) self-attention to learn dense local dependencies. To this end, consider an autoregressive task over sequences of length $n$, where the prediction of the next token depends only on the immediately preceding $l$ tokens, with $l \ll n$. For this task, the ideal attention matrix $P \in \mathbb{R}^{n \times n}$ would satisfy $P[i, j] > 0$ for $0 \leq i - j \leq l$, and $P[i, j] = 0$ otherwise.

When attempting to learn this dependency pattern using full causal attention, the model aims to approximate a matrix $P'$ using the self-attention operation such that $P'[i, j] = P[i, j]$ for $i - j \geq 0$, and treats $P'[i, j]$ as a "don't care" term for $i - j < 0$ (since these terms are masked in causal attention). An illustration of such an attention pattern is shown in Figure 2a, where $n = 6$ and $l = 2$; red entries represent masked (don't care) terms. Importantly, $P'$ is a matrix of rank $n$, which grows linearly with the sequence length. During training, the attention mechanism attempts to replicate $P'$ using softmax$((QK^T + M_{\mathcal{S}})/\sqrt{d})$, where $Q = [\mathbf{q}_0, \ldots, \mathbf{q}_{n-1}]^T \in \mathbb{R}^{n \times d}$, $K = [\mathbf{k}_0, \ldots, \mathbf{k}_{n-1}]^T \in \mathbb{R}^{n \times d}$ be the query and key matrices and $M_{\mathcal{S}} \in \mathbb{R}^{n \times n}$ be the causal mask (i.e., $M_{\mathcal{S}}[i, j] = 0$ for $i - j \geq 0$ and $-\infty$ otherwise). However doing so requires representing $O(n^2)$ non-masked entries in $P'$ using only $Q$ and $K$ of $O(nd)$ dimension. This mismatch becomes a critical bottleneck in settings where $n \gg d$, leading to logit explosion and training instability.

A sliding window local attention does not suffer from the same limitations when capturing such local dependencies. It attempts to reconstruct the ideal attention matrix $P$ only for the subset of entries $\{(i, j) : 0 \leq i - j \leq l'\}$, where the local attention span $l' \ll n$ and is on the same order as $l$. An example of an attention pattern learned by a sliding window local attention is shown in Figure 2b. In this case, the attention mechanism needs to learn only $O(nl')$ entries, which is significantly smaller than $O(n^2)$ for full attention. As a result, local attention is better suited for learning dense local dependencies compared to full attention.

### 3.1 VALIDATION THROUGH A SYNTHETIC TASK

Our synthetic task is designed to evaluate the representational power of the self-attention, $Y = $ softmax$((QK^T + M_{\mathcal{S}})/\sqrt{d})V$ (where $V = [\mathbf{v}_0, \ldots, \mathbf{v}_{n-1}]^T \in \mathbb{R}^{n \times d}$ be the value matrix), in capturing local dependencies when $Q$ and $K$ are allowed to freely take any values. The goal is to predict the output $Y = [\mathbf{y}_0, \ldots, \mathbf{y}_{n-1}]^T \in \mathbb{R}^{n \times d}$ of a sequence given the input $V$, such that $Y$

satisfies $Y = PV$ for a ground truth attention matrix $P \in \mathbb{R}^{n \times n}$. The matrix $P$ is constructed to encode dense local dependencies, typically as a banded matrix where only entries within a fixed window $l$ around the diagonal can be non-zero. Therefore, predicting $Y$ from $V$ using an attention mechanism effectively requires learning $Q$ and $K$ such that $P \approx \mathrm{softmax}((QK^T + M_S)/\sqrt{d})$ is satisfied, where $M_S$ denotes the appropriate masking matrix for full and local attention.

To that end, we generated a $2500 \times 2500$ ground truth attention matrix $P$ such that

$$P[i,j] = \begin{cases} p_{ij}, & \text{if } 0 < i - j \leq 50 \\ 0, & \text{otherwise} \end{cases}$$

where each $p_{ij}$ is independently drawn from a Bernoulli distribution with probability $0.5$. The matrix $P$ is then row-normalized to ensure it represents a valid attention distribution. We set $V$ to be the identity matrix of size $2500 \times 2500$, so that each $\mathbf{y}_i$ can be expressed as a unique linear combination of the $\mathbf{v}_j$s. This setup guarantees the uniqueness of $P$ in the relation $Y = PV$.

We trained both full and local self-attention operations for $100K$ steps using the Adam optimizer, with the key/query dimensionality set to $25$. For the local attention, we used a sliding window of span $50$. The training losses for both models are shown in Figure 2c. As illustrated, local attention leads to faster convergence and achieves significantly lower training loss compared to full attention after $100K$ steps, indicating its superior ability to model dense local dependencies. Additionally, we plot the maximum logit values of both self-attention mechanisms against their corresponding training losses, as shown in Figure 2d. The figure shows that local attention attains the same training loss with significantly smaller logits, underscoring full attention's greater susceptibility to the logit explosion problem when modeling local dependencies.

## 4 Long Short-attention: The Solution to Logit Explosion

Since local attention mechanisms are better suited than full attention for capturing dense local dependencies and induce less logit explosion, we propose the use of dedicated local attention heads to explicitly capture short-range interactions. However, local attention alone cannot capture the long-range dependencies. To overcome this limitation, our approach integrates local and full attentions, thereby enabling the joint modeling of both short- and long-range dependencies. We rely on the assumption that the overall attention matrix $P$ can be approximately decomposed as $P \approx P_{S_0} + \cdots + P_{S_{H_s-1}} + P_{L_0} + \cdots + P_{L_{H_l-1}}$ where each $P_{S_i}$ captures local dependencies within a small attention span $p \ll n$, and each $P_{L_j}$ captures long-range dependencies. Each $P_{L_j}$ is assumed to be low-rank. This assumption is motivated by the observation that, in many applications, only a small number of "keyword" tokens receive attention in long-range interactions, resulting in low-rank long-range attention patterns.

Given such a decomposition, the attention output can be approximated as:

$$Y = PV \approx \sum_{i=0}^{H_s-1} P_{S_i} V + \sum_{i=0}^{H_l-1} P_{L_i} V$$

$$\approx \sum_{i=0}^{H_s-1} \mathrm{softmax}\left(\left(Q_{S_i} K_{S_i}^T + M_s\right)/\sqrt{d}\right) V + \sum_{i=0}^{H_l-1} \mathrm{softmax}\left(\left(Q_{L_i} K_{L_i}^T + M_l\right)/\sqrt{d}\right) V$$

where $M_s$ and $M_l$ are the attention masks for short-range and long-range attention, respectively. In practice, we implement this combined mechanism using a $(s + l)$-head attention module, referred to as Long Short-attention (LS-attention), with $s$ short-range (local) attention heads and $l$ long-range (full) attention heads. Therefore, the output of LS-attention is given by:

$$\mathrm{LS\text{-}attn}(X) = \mathrm{Concat}(O^{(0)}, \ldots, O^{(H-1)})W_O, \text{ such that}$$

$$O^{(i)} = \mathrm{softmax}\left((Q^{(i)} K^{(i)^T} + M^{(i)})/\sqrt{d}\right) V^{(i)} = \mathrm{softmax}\left((XW_Q^{(i)} W_K^{(i)^T} X^T + M^{(i)})/\sqrt{d}\right) XW_V^{(i)}$$

where $H = s + l$, $O^{(i)}$ is the output of the $i$-th attention head, $M^{(i)}$ is the attention mask matrix for the $i$-th attention, and set to local attention mask for the first $s$ heads and to the full attention

mask (such as causal attention) for the last $l$ heads. In practice, we do not implement the LS-attention using the above parallel form. Rather, we use the efficient self-attention implementation of Dao et al. (2022); Dao (2024); Shah et al. (2024).

### RUNTIME AND MEMORY REQUIREMENTS

A full attention head requires $O(n^2d)$ FLOPs. In contrast, a local attention head with an attention span of $p$ requires only $O(npd)$ FLOPs. Therefore, an LS-attention module with $s$ local heads and $l$ full heads requires approximately $O(n(sp + nl)d) \approx O(n^2ld)$ FLOPs, assuming $p \ll n$. In comparison, a vanilla $(s + l)$-head attention requires $O((s + l)n^2d)$ FLOPs, which is roughly $(s + l)/l$ times more than LS-attention.

During inference in a transformer model with auto-regressive generation, the KV-cache (Pope et al. (2023); Zhang et al. (2023)) is used to store the key and value vectors of previous tokens to compute the attention scores for the future queries in the MHSA operation. The size of the KV-cache for a full attention head grows linearly with sequence length. In contrast, it remains nearly constant for a local attention head. Therefore, if the total number of attention heads remains the same, LS-attention reduces the KV-cache size by a factor of approximately $(s+l)/l$ compared to MHSA during long-sequence generation.

## 5 EXPERIMENTAL RESULTS AND ANALYSIS

This section examines how sequence length affects logit explosion and training stability in transformer models for natural language and speech processing tasks, where local dependencies are typically dense. We further demonstrate that the composed attention mechanism, LS-Attention, mitigates logit explosion and training instability. Alternative structured self-attention mechanisms and other training stabilization methods are also evaluated on their applicability in addressing long-sequence training instability.

### 5.1 EXPERIMENTAL SETUP

**Model Architecture** For most experiments, we used the GPT-2 Small model with 12 layers, an embedding dimension ($d$) of 768, 12 attention heads, and a feedforward dimension of 3072. As the baseline multi-head self-attention (MHSA), we employed the CUDA implementation of Flash-attention, specifically Flash-attention-2 from Dao (2024).

**Hyperparameters of LS-Attention** In experiments with LS-attention, we replaced the MHSA module with LS-attention. For an $H$-head LS-attention configuration, we used full (long-range) attention in one head, and local (short-range) attention in remaining $H - 1$ heads. The attention span for each local head was fixed at 100.

**Training Details** We trained all models using the AdamW optimizer with a weight decay of $1\mathrm{e}{-1}$, $\beta_1 = 0.9$, and $\beta_2 = 0.95$. Gradient clipping was applied with a maximum norm of $1.0$. The learning rate followed a cosine decay schedule with linear warmup: the maximum learning rate was set to $6\mathrm{e}{-4}$, the minimum to $6\mathrm{e}{-5}$, with $2K$ warmup steps and a total of $600K$ decay steps. Across all experiments, we fixed the total number of tokens per batch to $2^{19}$. Consequently, when using longer sequence lengths, we proportionally reduced the number of sequences per batch to maintain a constant token budget. Unless stated otherwise, we used mixed-precision training with the bfloat16 (BF16) data type.

### 5.2 EXPERIMENTAL RESULTS ON NATURAL LANGUAGE DATA

**Dataset and Preprocessing** We conducted our experiments on the PG-19 dataset (Rae et al. (2020)), a collection of English-language books. All texts were normalized using NMT_NFKC and tokenized with a SentencePiece unigram model with a vocabulary size of $10K$.

**Results** To investigate the effect of sequence length on training stability, we trained the above mentioned autoregressive baseline transformer (using Flash-attention as MHSA) for different sequence

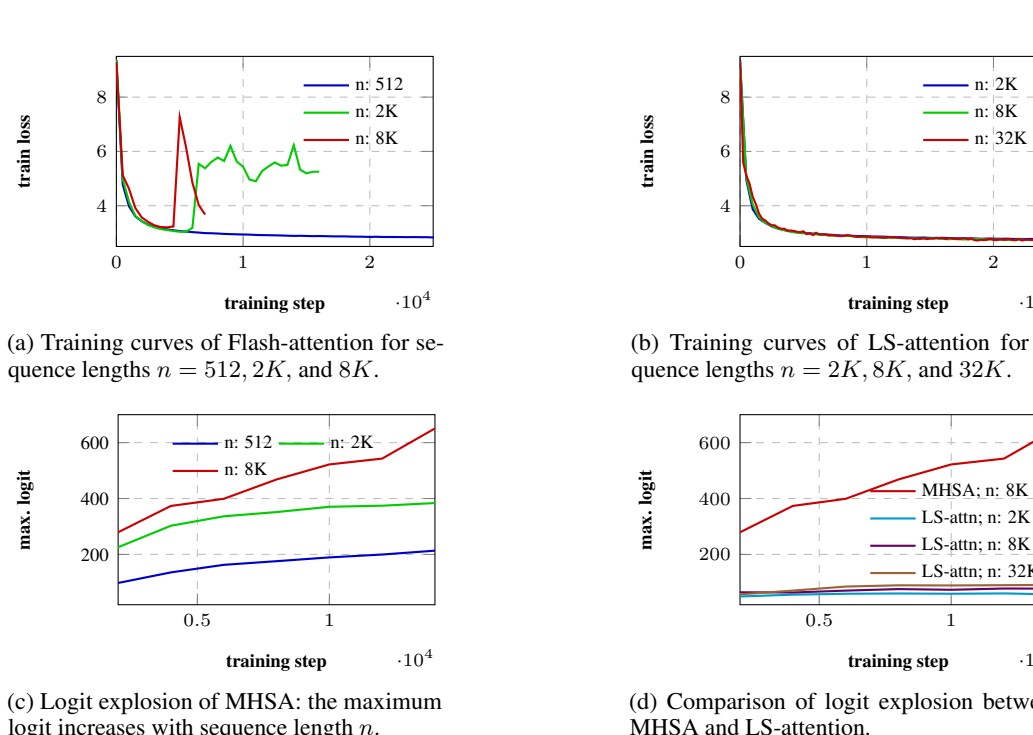

(a) Training curves of Flash-attention for sequence lengths $n = 512, 2K$, and $8K$.

(b) Training curves of LS-attention for sequence lengths $n = 2K, 8K$, and $32K$.

(c) Logit explosion of MHSA: the maximum logit increases with sequence length $n$.

(d) Comparison of logit explosion between MHSA and LS-attention.

Figure 3: Training stability and logit explosion on the PG-19 dataset (Rae et al. (2020)).

lengths. Figure 3a plots the training curves for sequence lengths of $512, 2K$, and $8K$. For the shorter sequence length ($n = 512$), the model trains stably, with the loss monotonically decreasing over the first $25K$ steps. However, when the sequence length is increased to $n = 2K$ and $n = 8K$, the training becomes unstable. In these cases, the loss initially decreases but then suddenly diverges, confirming that longer sequences can induce training instability. To further analyze this phenomenon, we tracked the maximum absolute logit values of the MHSA layes during training. As shown in Figure 3c, the maximum logit grows more rapidly with longer sequences. This result suggests that longer sequences cause greater logit explosion, which in turn contributes to the observed instability.

Next, we trained our baseline model by replacing the Flash-attention layers with LS-attention (comprising one full attention head and 11 local heads) for sequence lengths $n = 2K, 8K$, and $32K$. Figure 3b presents the training curves for these runs. As shown in the figure, the model with LS-attention does not exhibit any training instability during the first $25K$ training steps. To evaluate whether this improved training stability corresponds to mitigation of logit explosion, we compared the maximum absolute logit values of LS-attention and Flash-attention in Figure 3d. The figure clearly demonstrates that LS-attention substantially reduces the maximum logit values to negligible levels compared to Flash-attention, indicating that LS-attention effectively addresses logit explosion.

We provide additional results on the English split of the Wiki40B dataset (Dao et al. (2022)) in Appendix B. These results further confirm the above observations.

### 5.3 EXPERIMENTAL RESULTS ON SPEECH DATA

**Dataset and Preprocessing** To validate our observations on a speech dataset, we used the $6K$-hour unlabeled subset of the LibriLight corpus (Kahn et al. (2020)). The audio data was tokenized using a pretrained HuBERT model (Hsu et al. (2021)) with a $500$-cluster K-means tokenizer, resulting in a vocabulary size of $500$. Our decoder only transformer models are autoregressively trained on the tokenized dataset.

**Results** Following the same protocol as with the PG-19 dataset, we trained the baseline model (using Flash-attention as the MHSA layers) on various sequence lengths. The training curves are

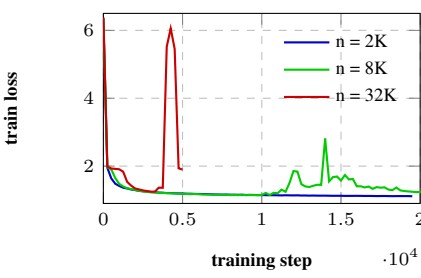
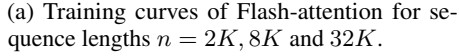
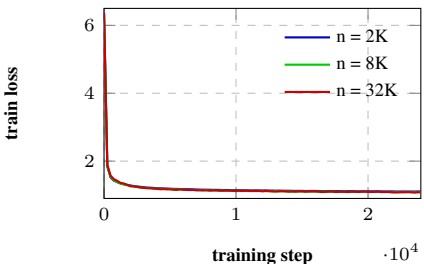

(a) Training curves of Flash-attention for sequence lengths $n = 2K, 8K$ and $32K$.

(b) Training curves of LS-attention for sequence lengths $n = 2K, 8K$ and $32K$.

Figure 4: Training stability evaluation on the LibriLight speech dataset ($6K$ split) Kahn et al. (2020).

plotted in Figure 4a. While the model trains stably with a sequence length of $2K$, it exhibits instability at longer lengths of $8K$ and $32K$. For these longer sequence lengths, the training loss diverges after an initial monotonic decrease. This instability correlates with logit explosion; Figure 8a in Appendix C shows that maximum absolute logit values increase more rapidly as sequence length grows.

We then repeated these experiments, replacing MHSA modules with LS-attention. As shown in Figure 4b, the model now trains stably even on sequences as long as $32K$. Furthermore, Figure 8b confirms that LS-attention mitigates the logit explosion seen in the baseline. These results indicate that LS-attention, which combines both local and full attention, effectively addresses the training instability encountered during long-sequence training.

## 5.4 RESULTS WITH LARGER MODEL

To assess whether the previous findings generalize to larger transformer models, we trained the **GPT-2 Large model** with 36 layers, 20 heads, embedding dimension 1280, and feedforward dimension 5120. The resulting model has approximately $750M$ parameters and was trained using the same setup described in Section 5.1. Figure 5 compares the training curves of the model using Flash-attention and LS-attention at a sequence length of $8K$. As before, the model with Flash-attention exhibits training instability, while LS-attention enables stable training.

## 5.5 COMPARISON OF INFERENCE TIME

We evaluated the inference latency of the LS-attention against the baseline using Flash-attention. Both models were benchmarked on an Nvidia A40 GPU with the BF16 data type. Latency was measured during a single forward pass over a batch of input sequences. The results, presented in Table 1, show that LS-sttention significantly reduces inference time on longer sequence lengths. At a sequence length of $n = 32K$, LS-attention is nearly $44\%$ faster than the Flash-attention baseline. In general, as the sequence length increases, the time reduction achieved by LS-attention is expected to asymptotically approach a factor of $H/l$, where $H$ denotes the total number of heads in Flash-attention and $l$ represents the number of full-attention heads in LS-attention.

Figure 5: Comparison of training stability using larger model ($\approx 750M$ Parameters) on PG-19 dataset for sequence length $n = 8K$.

## 5.6 ADDITIONAL OBSERVATIONS

**Does Other Structured Self-attention Mitigates Long-Sequence Training Instability?** Other structured self-attention mechanisms with local attention heads can also mitigate long-sequence train-

| Seq. len ($n$) | Attention Type | | Reduction |
|---|---|---|---|
| | Flash-attn | LS-attn | |
| $8K$ | 52.50 | 48.75 | 7.14% |
| $32K$ | 374.00 | 210 | 43.85% |

Table 1: Reduction in inference time (in milliseconds per sequence) using LS-attention on Nvidia A40 GPU.

ing instability, as shown in Appendix D. However, while LS-attention can be efficiently implemented using publicly available software packages such as Flash Attention Dao et al. (2022); Dao (2024) and xFormers Lefaudeux et al. (2022), other attention mechanisms such as Guo et al. (2019) require hardware-specific custom implementations.

**Does Existing Training Stabilation Methods Mitigates Long-Sequence Training Instability?**
In Appendix E, we evaluated three existing training stabilization methods: (1) QK-normalization (Henry et al. (2020)), (2) Z-loss (Chowdhery et al. (2023)), and (3) the AdaGC optimizer (Wang et al. (2025)). We found that Z-loss and AdaGC failed to mitigate long-sequence training instability, whereas QK-normalization led to significantly slower convergence than LS-attention – yielding more than 30% higher perplexity over the same number of training steps.

**Can Flash-attention with Full Precision Training Mitigates Long-Sequence Training Instability?**
As discussed in Appendix F, Flash-attention with full FP32 precision training can mitigate long-sequence training instability. However, it requires over 15 times as many GPU hours to reach the same training loss as LS-attention.

**How Sensitive Is LS-attention Performance to the Number of Local and Full Attention Heads?**
In our analysis in Appendix G, we find that LS-attention is only mildly sensitive to the number of both local and full attention heads, provided that at least one of each is present.

## 6 CONCLUSION

This paper identifies a key source of training instability in transformer models: the self-attention's limited ability to capture dense local dependencies. This limitation causes the logits of self-attention to explode when training on long sequences, leading to instability. To address this, we propose using a combination of local and full attention heads, where the local heads capture short-range dependencies and the full heads capture long-range dependencies. This composed self-attention, referred to as Long Short-attention (LS-attention), mitigates logit explosion and instability during long-sequence training. A wide range of experiments demonstrates that long-sequence training leads to logit explosion in standard multi-head self-attention, while LS-attention mitigates it, resulting in stable training. Furthermore, LS-Attention improves transformer efficiency compared to state-of-the-art multi-head self-attention implementations such as Flash-attention.

## 7 BROADER IMPACT

Training instability in Transformer models poses a serious bottleneck in large-scale AI development, with substantial financial, operational, and environmental consequences. In both industrial and academic settings, a single failed training run – often occurring after days or weeks of computation – can result in thousands of wasted GPU-hours, escalating costs and carbon emissions. To mitigate this risk, practitioners resort to engineering-intensive workarounds such as curriculum-based sequence scaling and manual monitoring, further inflating resource overhead. By addressing a core source of instability, our approach offers a scalable and efficient path toward training robust, long-sequence Transformer models.

## 8 REPRODUCIBILITY STATEMENT

Our source code is included in the supplementary materials.

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

## A    USE OF LARGE LANGUAGE MODELS

Large Language Models (LLMs) were used to improve textual clarity and to facilitate comprehensive information retrieval on a wide range of subjects from online resources.

## B    RESULTS ON WIKI40B (ENGLISH SPLIT) DATASET

**Dataset and Preprocessing**    To further validate our findings, we conducted experiments on the English split of Wiki40B (Guo et al. (2020)) dataset. This dataset is a cleaned version of Wikipedia designed for large-scale NLP tasks. Following the same preprocessing steps used for the PG-19 dataset, we normalized the texts with NMT_NFKC and tokenized them using a SentencePiece unigram model with a vocabulary size of $10K$.

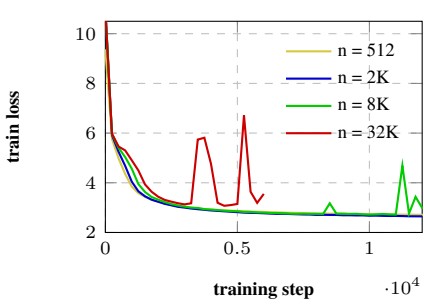

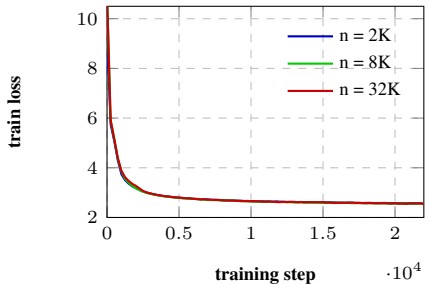

(a) Training curves of Flash-attention for sequence lengths $n = 512, 2K, 8K$ and $32K$.

(b) Training curves of LS-attention for sequence lengths $n = 2K, 8K$, and $32K$.

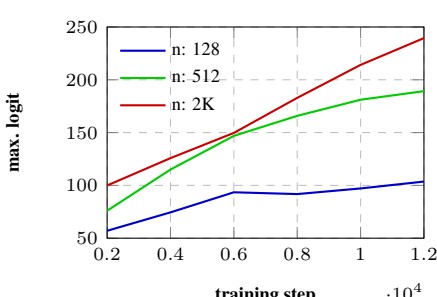

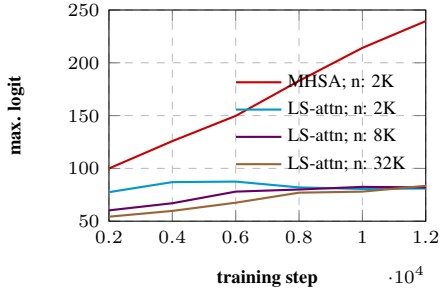

(c) Logit explosion of MHSA: the maximum logit increases with sequence length $n$.

(d) Comparison of logit explosion between MHSA and LS-attention.

Figure 6: Training stability and logit explosion at long sequence lengths on the Wiki40b (English) dataset (Guo et al. (2020)).

**Results** As before, we trained the baseline transformer (using Flash-attention as the MHSA) with various sequence lengths. The training curves, shown in Figure 6a, mirror our previous findings. For shorter sequences ($512$ and $2K$), the model trains stably, with the loss decreasing monotonically over the first $15K$ steps. However, at longer sequence lengths ($8K$ and $32K$), the training becomes unstable and the loss diverges after an initial decrease. This instability again corresponds to a growth in logit values of MHSA. Figure 6c, which plots the maximum absolute logits of the MHSA layers, confirms that logit explosion is magnified by longer sequences, suggesting it is a primary cause of the training instability.

We then replaced the Flash-attention layers with the LS-attention and repeated the training for sequence lengths of $n = 2K, 8K$, and $32K$. The training curves in Figure 6b show that the model now trains stably across all tested lengths for the first $25K$ steps. Figure 6d compares LS-attention and Flash-attention with respect to logit explosion, confirming once again that the training stability achieved by LS-attention corresponds to effective mitigation of logit explosion.

## C  LOGIT EXPLOSION ANALYSIS FOR LIBRILIGHT DATASET

Building on the instability results for the LibriLight dataset (Section 5.3), we now analyze the underlying logit behavior. Figure 8a plots the maximum absolute logit values of the MHSA layers in the baseline transformer model. Consistent with our findings on other datasets, the logit values grow more rapidly as sequence length increases, reinforcing the link between sequence length and logit explosion. When compared to this baseline, LS-attention again proved to be effective. As shown in Figure 8b, LS-attention successfully mitigates the logit explosion observed in MHSA, reaffirming the trend seen across all the three datasets.

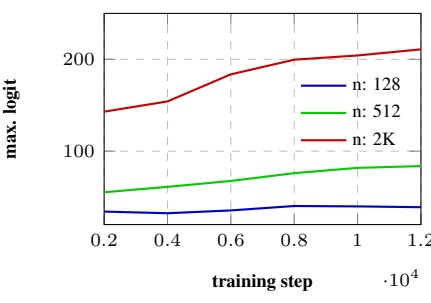

(a) Logit explosion of MHSA: the maximum logit increases with sequence length $n$.

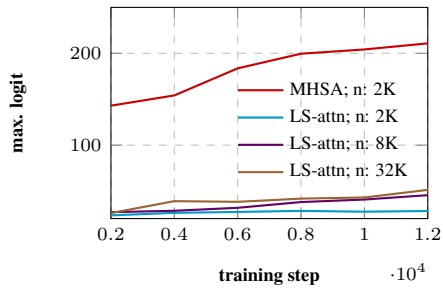

(b) Comparison of logit explosion between MHSA and LS-attention.

Figure 8: Logit explosion at longer sequence lengths on LibriLight speech dataset (6K split).

## D  EVALUATION OF ALTERNATIVE STRUCTURED SELF-ATTENTIONS FOR LONG SEQUENCE TRAINING

Several existing works (Child et al. (2019); Beltagy et al. (2020); Zaheer et al. (2020); Jiang et al. (2024); Guo et al. (2019)) have explored various structured self-attention mechanisms, including sparse attentions, to improve the computational efficiency of transformers on long sequences. Most of these structured self-attention methods also involve local attention and therefore should be able to stabilize long-sequence training. To verify this, we trained our baseline model with the MHSA layer replaced by the structured self-attention of Guo et al. (2019) on the PG-19 dataset with $n = 2K$. As expected, it was able to stabilize training. However, we found that the PyTorch implementation of this attention is nearly $2\times$ slower for sequence lengths below $8K$ and almost $10\times$ slower for sequence lengths of $32K$ compared to LS-Attention. The main advantage of LS-attention over the attention of Guo et al. (2019) is that LS-attention can be easily implemented using freely available packages such as Flash attention (Dao et al. (2022); Dao (2024)) and xFormers (Lefaudeux et al. (2022)). In contrast, optimized attention implementations, such as those in Guo et al. (2019), require hardware-specific programming.

## E  EVALUATION OF EXISTING ALTERNATIVE TRAINING STABILIZATION METHODS

In this section, we evaluate whether existing alternative stabilization techniques can improve long-sequence training. To this end, we assessed three methods: (1) QK-normalization (Henry et al. (2020)), (2) Z-loss (Chowdhery et al. (2023)), and (3) the AdaGC optimizer (Wang et al. (2025)).

The learning curves for these methods, obtained by training our baseline model on the PG-19 dataset with a sequence length of $n = 2K$, are shown in Figure 9a. The results indicate that Z-loss and the AdaGC optimizer failed to stabilize training. Although QK-normalization successfully stabilized training, it converged too slowly compared to LS-attention, reaching a training loss of approximately 3.16 (i.e., perplixity 23.57) over the first $25K$ steps. In contrast, LS-attention achieved a substantially lower loss of about

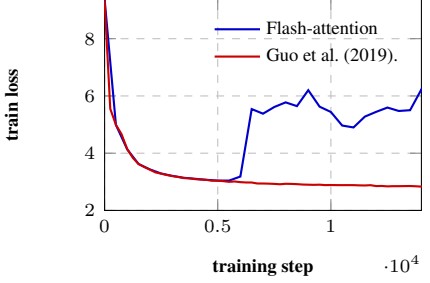

Figure 7: Evaluation of training stability by the structured attention of Guo et al. (2019) on PG-19 dataset with $n = 2K$.

2.77 (i.e., perplexity 15.96) over the same period, corresponding to roughly a $32\%$ more reduction in perplexity.

# F  COMPARISON OF LS-ATTENTION (BF16) AND FLASHATTENTION (FP32)

In this section, we investigate whether training Flash-attention with full FP32 precision, rather than the default mixed precision with BF16, can stabilize long-sequence training. Prior work, such as Golden et al. (2024), has noted that Flash-attention – the efficient MHSA implementation – is particularly vulnerable to numerical instability caused by the reduced precision of low-bit datatypes. We therefore explored full-precision training of Flash-attention as a potential stabilization strategy.

Our experimental results suggest that full-precision training can stabilize long-sequence training, though it incurs higher computational cost (both in terms of the number of training steps and GPU hours). Figure 9b compares the GPU hours required by LS-attention and full-precision Flash-attention to train the baseline model on the PG-19 dataset with $n = 8K$. The findings show that Flash-attention with full-precision training may require over 15 times more GPU hours to achieve comparable perplexity to LS-attention.

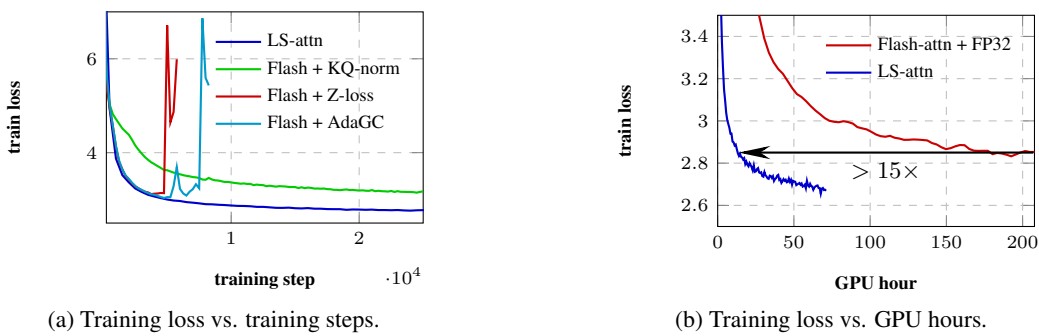

(a) Training loss vs. training steps.  (b) Training loss vs. GPU hours.

Figure 9: (a) Evaluation of alternative training stabilization methods for stabiling long sequence training. The following methods have been explored: (1) Flash-attention using QK-normalization, (2) Flash-attention with Z-loss, and (3) Flash-attention with AdaGC optimization. (b) Comparison of LS-attention and Flash-attention using full FP32 precision training. All the experiments have been conducted on PG-19 dataset.

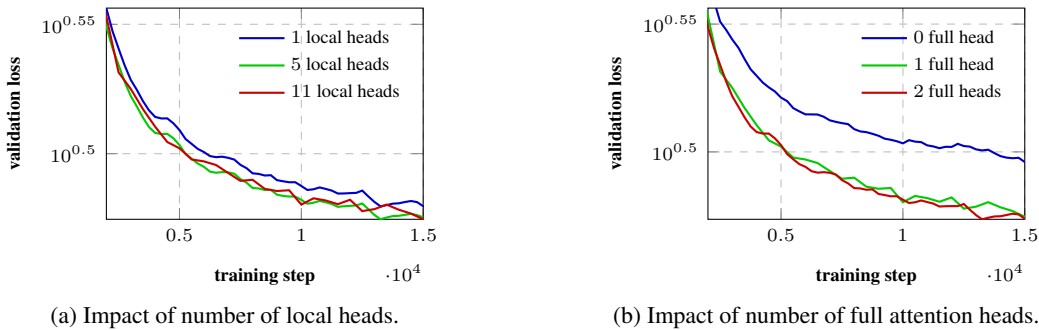

(a) Impact of number of local heads.  (b) Impact of number of full attention heads.

Figure 10: Effect of the number of local and full attention heads in LS-attention. All experiments were conducted on the PG-19 dataset with $n = 8K$.

# G  SENSITIVITY ANALYSIS OF HYPERPARAMETERS

In this section, we analyze the sensitivity of LS-attention's performance to the number of local and full attention heads.

## G.1 SENSITIVITY ANALYSIS OF THE NUMBER OF LOCAL ATTENTION HEADS

To investigate the effect of the number of local attention heads, we fixed the number of full-attention heads to 1 and varied the number of local attention heads to 1, 5, and 11. Note that setting the number of local attention heads below 11 while keeping one full-attention head reduces the total number of attention heads from the default value of 12, thereby decreasing the number of parameters in the self-attention layers. To compensate for this reduction, we increased the feed-forward dimension appropriately so that the total number of model parameters remained constant.

The learning curves for different numbers of local attention heads are shown in Figure 10a. The results suggest that varying the number of local attention heads has little effect on the convergence rate, with the configuration using a single local-attention head exhibiting slightly slower convergence.

## G.2 SENSITIVITY ANALYSIS OF THE NUMBER OF FULL ATTENTION HEADS

To investigate the effect of the number of full-attention heads in LS-attention, we conducted experiments with the number of full-attention heads set to 0 (i.e., no full attention), 1, and 2, while keeping the total number of attention heads fixed at 12. The learning curves for these three settings are shown in Figure 10b. The figure suggests that increasing the number of full-attention heads from 0 to 1 yields a substantial performance improvement: the model with 0 full-attention heads reached a validation loss of 3.14 after $15K$ steps, whereas the model with 1 full-attention head achieved 2.98 over the same period. This improvement indicates that adding a single full-attention head helps the model capture long-range dependencies, thereby improving performance compared to the configuration without full attention. However, increasing the number of full-attention heads from 1 to 2 did not provide further benefits, suggesting that only a small number of full-attention heads is adequate for modeling long-range dependencies.

