# OpenReview forum: "Transformer Instability in Long Sequence Training: The Underestimated Role of Short-Range Dependencies"
_ICLR.cc/2026/Conference — Submitted to ICLR 2026_

### Official Review · Reviewer_DTqn · 2025-10-26

**Soundness:** 1
**Presentation:** 1
**Contribution:** 1
**Rating:** 0
**Confidence:** 5

**Summary:**

This work proposes replacing standard attention heads in a transformer with local-window ones to enhance training stability. Numerous training curves are presented.

**Strengths:**

The reviewer was unable to identify any strengths in this submission.

**Weaknesses:**

The quality of this submission is well below acceptable standards on all fronts and fails to meet even the most basic expectations:

* Method: the method proposed---local attention is not novel at all. The use of local attention typically leads to performance degradation, which is not discussed in the paper.

* Experiments: several training loss plots are reported, but no test performance is included---this omission alone is a valid reason for strong rejection)---we expect both perplexity and downstream task performance to properly evaluate language models, and downstream tasks for speech models. In addition, the training curves are shown only for a single run, which appears to assume that training instability systematically affects the baseline model--- this seems unlikely: more runs need to be conducted to confirm the instability trend and support the claim.

* Presentation: The writing is overall poor. The authors appear to conflate transformer "language models" with the transformer "architecture" in both the abstract and introduction. A synthetic task is introduced in Sec. 3 before the experimental section, and an excessive amount of space is devoted to describing basic concepts (e.g., the equations in Sec. 4).

The reviewer recommends that the authors consult recently published ICLR papers to better understand the expected level of rigor and presentation quality.

**Questions:**

The reviewer has no further questions and considers it unlikely that this work will become acceptable after any rebuttal or discussion.

---

### Official Review · Reviewer_YFyy · 2025-10-30

**Soundness:** 2
**Presentation:** 2
**Contribution:** 2
**Rating:** 2
**Confidence:** 3

**Summary:**

This paper investigates the training instability of Transformers when handling long sequences, authors proposed: self-attention's limited capacity to capture short-range dependencies, which is particularly problematic in language modeling where tokens heavily depend on immediate neighbors. The authors demonstrate that this limitation leads to logit explosion during long-sequence training, causing training divergence.

They propose Long Short-attention, which combines local attention heads with full attention heads. The local heads explicitly capture short-range dependencies while the full heads maintain long-range context. The authors show that LS-attention effectively prevents logit explosion, stabilizes training, and improves inference efficiency compared to state-of-the-art MHSA implementations.

**Strengths:**

The paper's primary strength is its clear identification of a fundamental issue in Transformer training that has been overlooked in prior work. The synthetic experiment in Section 3.1 provides compelling evidence for the core hypothesis. The proposed LS-attention solution is elegant, simple to implement, and brings dual benefits, training stability and computational efficiency. The empirical validation is thorough, covering multiple modalities and model scales. The comparison with existing stabilization methods and the demonstration of 44% inference speedup make a strong practical case. The work's potential impact on reducing failed training runs and associated computational costs is substantial.

**Weaknesses:**

The paper would benefit from a more detailed comparison with other structured attention approaches beyond the brief mention in Section 5.6. While the authors note that LS-attention is more efficient than methods like Guo et al. (2019), a more systematic comparison of stability and efficiency across different long-sequence attention mechanisms would strengthen the paper.

The choice of using just one full attention head appears arbitrary, more analysis of the optimal ratio between local and full heads for different tasks would be valuable. Additionally, the paper focuses exclusively on causal (autoregressive) attention; extending the analysis to bidirectional attention patterns would broaden the applicability. The attention span of 100 for local heads is fixed across experiments without justification. The authors don't explore how this parameter affects performance or whether it should vary across different tasks or modalities. A more thorough analysis of the optimal attention span would strengthen the practical guidance. The comparison with existing stabilization methods (QK-normalization, Z-loss, AdaGC) is limited to a single dataset (PG-19) and doesn't explore how these methods perform on more challenging benchmarks or longer sequences.

The formatting of the paper could be largely improved (e.g. out-of-line character ‘t’ on line 063, spaces were not compactly used).

**Questions:**

The paper consistently uses one full attention head with eleven local heads. What empirical evidence supports this specific ratio? Would this optimal ratio change for different tasks or sequence lengths, and how should practitioners determine the right balance?

While the paper demonstrates stability benefits for autoregressive models, how would LS-attention perform in bidirectional settings like BERT? Are there any modifications needed for non-causal attention patterns?

The attention span of 100 for local heads is fixed across experiments. What informed this choice, and how sensitive is performance to this parameter? Would different modalities (e.g., speech vs. text) benefit from different window sizes?

---

### Official Review · Reviewer_fJKX · 2025-11-02

**Soundness:** 4
**Presentation:** 4
**Contribution:** 2
**Rating:** 4
**Confidence:** 4

**Summary:**

This paper investigates the root cause of training instability in long-sequence transformer models and attributes it to self-attention’s inability to effectively capture dense short-range dependencies. The authors propose Long Short-attention (LS-attention), a hybrid mechanism combining local (short-range) and full (global) attention heads to stabilize optimization and mitigate logit explosion. Experiment results on language tasks show smoother training loss, reduced logit magnitudes, and improved computational efficiency compared to FlashAttention and other baselines. However, the paper does not report final accuracy or downstream performance, leaving it unclear whether the improved stability translates into comparable or superior model quality.

**Strengths:**

1. The paper provides a convincing diagnosis of instability, linking dense local dependencies to logit explosion through both intuitive reasoning and controlled experiments.
2. LS-attention is architecturally straightforward, compatible with existing implementations (e.g., FlashAttention), and easy to adopt in practice.
3. The authors studied a lot of related works and give concise comparisons. After studying those related works, they pinpoint the issue of existing methods and follow up with their proposed solution.

**Weaknesses:**

1. The study does not report whether LS-attention achieves comparable or better final performance than standard MHSA on any dataset. Improved training stability is demonstrated, but model effectiveness, in terms of accuracy, perplexity at convergence, or downstream task results, is not established.
2. Despite claims of general applicability across language, speech, and vision, the experiments are limited to text and speech; no quantitative results are provided for vision tasks.
3. The idea of hybrid attention structure is not new (e.g., Longformer, BigBird). Without stronger evidence of accuracy preservation or new capabilities, the contribution of this paper is considered incremental.

**Questions:**

1. You demonstrate improved training stability and reduced logit magnitudes, but do not report final validation or test accuracy/perplexity once training converges. Could you provide results showing whether LS-attention achieves comparable or superior final task performance relative to standard MHSA? Stability without quality preservation would limit practical impact.
2. The experiments are limited to text and speech datasets. Did you attempt to evaluate LS-attention on a vision benchmark (e.g., ImageNet, CIFAR, or ViT pretraining)? If not, what challenges prevented such evaluation?
3. Does introducing local attention heads reduce the model’s ability to capture long-range dependencies? Have you analyzed whether the local-global head ratio affects final accuracy or attention diversity, especially for tasks requiring global context?
4. How does LS-attention compare to existing hybrid attention architectures like Longformer or BigBird in both stability and final performance? Since those models already mix local and global patterns, a direct empirical comparison would clarify what distinguishes LS-attention.

---

### Official Review · Reviewer_rMWC · 2025-11-03

**Soundness:** 2
**Presentation:** 2
**Contribution:** 2
**Rating:** 2
**Confidence:** 4

**Summary:**

This paper investigates training instability in Transformers under long-sequence regimes and attributes the problem to dense short-range dependencies that overload the capacity of the global attention mechanism. The authors claim that this mismatch leads to “logit explosion” and unstable loss behavior, particularly when using FlashAttention for efficient computation.

To mitigate this, they propose Long-Short Attention (LS-Attention), which assigns most attention heads to local windows while keeping a few global heads for long-range dependencies. The method reportedly improves training stability and reduces compute cost on PG-19, Wiki40B, and LibriLight datasets.

**Strengths:**

The paper raises an important question regarding Transformer stability in long-sequence training.

**Weaknesses:**

1. The paper attributes training instability to “global attention under dense short-range dependencies” and specifically blames FlashAttention for numerical divergence (Fig 1). However, the implementation used in the paper is based on GPT-2 architecture with absolute sinusoidal position embeddings, not modern relative encodings (e.g., RoPE or ALiBi). It is well-known that absolute position embeddings cause variance drift in QK dot products for long sequences, which interacts poorly with blockwise softmax normalization in FlashAttention — leading to the observed “logit explosion”. Therefore, the instability is more plausibly a byproduct of the positional encoding choice, not a fundamental flaw of full attention or FlashAttention itself.

2. The claim that “FlashAttention exhibits training instability” is not supported by prior work. The only related paper, Is Flash Attention Stable?[1], shows higher numeric deviation at low precision (BF16) but no empirical evidence of loss divergence. The current paper overextends this result into a general claim of instability, without reproducing the experiment under alternative position encodings or precisions. This undermines the central hypothesis. Some recent works have successfully applied FlashAttention in model training. For instance, the Phi-3 Technical Report[6] explicitly states that the model was trained using a custom Triton kernel based on FlashAttention for both efficiency and stability.

3. The proposed LS-Attention is structurally equivalent to prior local + global attention models (Longformer[2]; BigBird [3]; MEGA [4]; Conformer [5]), differing only in motivation (“stability” vs. “efficiency”). The paper lacks any new theoretical insight or mechanism beyond reinterpreting known hybrids.

4. Furthermore, the claim that “FlashAttention exhibits training instability” is factually incorrect when viewed in the broader work. For instance, the Phi-3 Technical Report[6] explicitly states that the model was trained using a custom Triton kernel based on FlashAttention for both efficiency and stability.

[1] Is Flash Attention Stable? arXiv:2405.02803

[2] Longformer: The Long-Document Transformer. arXiv:2004.05150

[3] Big Bird: Transformers for Longer Sequences. arXiv:2007.14062

[4] Mega: Moving Average Equipped Gated Attention. arXiv:2209.10655

[5] Conformer: Convolution-augmented Transformer for Speech Recognition. arXiv:2005.08100

[6] Phi-3 Technical Report: A Highly Capable Language Model Locally on Your Phone. arXiv:2404.14219

**Questions:**

Some recommendations:

1. Re-run experiments using RoPE or ALiBi position encodings to verify whether instability persists.

2. Discuss relation to modern hybrid attention (Longformer, BigBird, MEGA, Conformer) more carefully

---

### Meta-Review · Area_Chair_gQ3h · 2026-01-06

**Summary:**

1. Causal claim may be flawed: The reported “FlashAttention instability/logit explosion” may be driven by the use of absolute sinusoidal positional embeddings (GPT-2 style) rather than an inherent issue with full attention or FlashAttention; reviewers ask to re-run with RoPE/ALiBi and different precisions.
2. Missing final quality metrics: The paper emphasizes stability curves but does not report final validation/test performance (e.g., perplexity/accuracy/downstream results), so it’s unclear whether stability comes without performance degradation.
3. Limited novelty / incremental contribution: LS-Attention looks very similar to prior local+global hybrid attention (e.g., Longformer, BigBird, MEGA, Conformer); stronger differentiation and direct comparisons are requested.
4. Insufficient ablations/sensitivity analysis: Key design choices (e.g., #global vs local heads, window size/span) appear arbitrary; reviewers want systematic sweeps and guidance.
5. Evidence scope and rigor: Claims of broad applicability (e.g., vision) lack quantitative results, and some experiments appear to rely on single runs rather than multiple seeds.
6. Presentation issues: Some reviewers note writing/organization/formatting problems that reduce clarity and perceived rigor.

**Reviewer Concerns:**

There is no rebuttal.

**Reviewer Scores:**

The reviewers won't change their scores because there is no rebuttal.

---

### Decision · Program_Chairs · 2026-01-26

Reject